# The Antiproliferative and Proapoptotic Effects of Cucurbitacin B on BPH-1 Cells via the p53/MDM2 Axis

**DOI:** 10.3390/ijms25010442

**Published:** 2023-12-28

**Authors:** Ping Zhou, Sisi Huang, Congcong Shao, Dongyan Huang, Yingyi Hu, Xin Su, Rongfu Yang, Juan Jiang, Jianhui Wu

**Affiliations:** 1Shanghai Engineering Research Center of Reproductive Health Drug and Devices, NHC Key Laboratory of Reproduction Regulation, Shanghai Institute for Biomedical and Pharmaceutical Technologies, Pharmacy School, Fudan University, Shanghai 200237, China; zhouping19a@163.com (P.Z.); sisihuang666@163.com (S.H.); shaocongcongscc@163.com (C.S.); hdy043@163.com (D.H.); suxiaoxin1982@163.com (X.S.); yangrongfu82@163.com (R.Y.); jiangjuan06@126.com (J.J.); 2Department of Pharmacology & Toxicology, Shanghai Institute for Biomedical and Pharmaceutical Technologies, Shanghai 200032, China

**Keywords:** triterpenoid compound, cucurbitacin B, prostatic hyperplasia, p53 signaling pathway

## Abstract

Cucurbitacin B (Cu B), a triterpenoid compound, has anti-inflammatory and antioxidant activities. Most studies only focus on the hepatoprotective activity of Cu B, and little effort has been geared toward exploring the effect of Cu B on the prostate. Our study identified that Cu B inhibited the proliferation of the benign prostatic hyperplasia epithelial cell line (BPH-1). At the molecular level, Cu B upregulated *MDM2* and thrombospondin 1 (*THBS1*) mRNA levels. Immunocytochemistry results revealed that the protein expressions of p53 and MDM2 were upregulated in BPH-1 cells. Furthermore, Cu B upregulated THBS1 expression and downregulated COX-2 expression in the BPH-1 cell supernatant. Altogether, Cu B may inhibit prostate cell proliferation by activating the p53/MDM2 signaling cascade and downregulating the COX-2 expression.

## 1. Introduction

The prostate is the major reproductive gland and affects male fertility and health. Microscopically, the prostate gland mainly consists of stromal and epithelium cells. The growth and development of the prostate are induced by stromal–epithelial interactions. The benign prostatic hyperplasia epithelial cell line (BPH-1) originated from human prostate tissue after transurethral resection and was first isolated and identified by S.W. Hayward et al. in 1994. The BPH-1 cell line lacks the expression of the androgen receptor [1]. Uncontrolled proliferation of BPH-1 cells can induce BPH and lead to bothersome complications, such as hematuria and renal insufficiency, significantly impairing a patient’s quality of life. BPH-1 cells stimulate aromatase expression in prostatic stromal cells (PrSCs) by producing prostaglandin E2 in a paracrine manner. Aromatase can convert testosterone into estradiol and have high expression in BPH patients [2]. As a well-established model for the human prostate biology of BPH, the BPH-1 cell line has been exploited to research anti-BPH candidates.

Natural products are compounds derived from animals, plant extracts, insects, marine organisms, and microorganisms. They include alkaloids, glycosides, sterols, flavonoids, polyphenols, triterpene glycosides, etc. Natural products have considerable economic value, and it has been reported that one-fourth of the world’s best-selling drugs derive from natural products and their derivatives [3]. These phytochemicals have remarkable pharmacological properties, such as anti-inflammatory, antioxidant, and immunomodulatory functions. Concurrently, phytochemicals can act on multiple therapeutic targets with fewer side effects [4]. Using natural product libraries, we screened a compound with potential anti-BPH activity. Cucurbitacin B (Cu B), a triterpenoid compound, is widely available in cucumber, pumpkin, and loofah. In general, triterpenoids exhibit potent antifungal, antibacterial, and hepatoprotective activities [5]. Comparative proteomics showed that the triterpenoid compound, luteolin, suppressed prostate cancer stemness through the induction of frizzled class receptor 6 (FZD6) expression [6]. Furthermore, cucurbitacin D exerted suppressive effects on cervical cancer by inhibiting phosphoinositide 3-kinase (PI3K)/protein kinase B (AKT) and the signal transducer and activator of transcription 3 (STAT3) signaling cascade and enhancing the expressions of tumor-suppressing genes such as *miR-145*, *miRNA-143*, and *miRNA34a* [7]. It is thus plausible that natural products—including triterpenoids—are believed to have vast potential in treating different diseases and deserve in-depth study.

Cucurbitacin B has anti-inflammatory and immunomodulatory activities, but studies have focused little on its roles in prostate cells and the potential molecular mechanism. In this study, the proliferative and apoptotic effects of Cu B on BPH-1 were measured using a Counting Kit-8 (CCK-8) assay and flow cytometry. Given the importance of the p53 signaling pathway in regulating cell proliferation and apoptosis, we investigated the gene and protein expressions of TP53, MDM2, and THBS1 using real-time quantitative polymerase chain reaction (RT-qPCR), immunocytochemistry, and enzyme-linked immunosorbent assay (ELISA). Our data demonstrated that Cu B acted as a proapoptotic agent to inhibit prostate cell proliferation by upregulating TP53, MDM2, and THBS1 expressions. In parallel, Cu B decreased COX-2 expression in BPH-1 cells. Altogether, we considered that Cu B treatment could inhibit prostate cell proliferation by stimulating the p53/MDM2 axis and downregulating COX-2 expression.

## 2. Results

### 2.1. Cu B Exhibited Antiproliferative Effects on BPH-1 Cells

The effects of Cu B on BPH-1 cells were evaluated using a CCK-8 assay. As Figure 1 shows, Cu B (12.5–200 nM) inhibited the growth of BPH-1 cells. Meanwhile, we selected doxazosin as the positive control to evaluate the antiproliferative activity difference between Cu B and doxazosin on prostate cells. We found that after treating cells with Cu B for 48 h, the inhibitory effect of Cu B (50–200 nM) on BPH-1 cell proliferation was slightly stronger than that of doxazosin (40 μM). Moreover, cytomorphological observation indicated that compared with the control group, Cu B treatment induced distinct morphological alterations, manifesting as cell shrinkage, rounding, and karyorrhexis (Figure 2).

### 2.2. Cu B Induced Apoptosis of BPH-1 Cells

In order to further evaluate the efficacy of Cu B on BPH-1 cells, we measured Cu B-treated cell apoptosis rates using flow cytometry. Annexin V-FITC/PI staining highlighted that after treatment with Cu B and Doxa, the apoptosis rates of BPH-1 cells (Figure 3) were drastically increased. The apoptosis index of BPH-1 cells in the high-dose Cu B group (50 nM and 100 nM) was remarkably higher than that in the control group.

### 2.3. Cu B Regulated Gene Levels of the p53/MDM2 Signaling Axis

Given the importance of *p53*-related genes on cell proliferation and apoptosis, we evaluated the relative mRNA levels of *p53* and its downstream target gene using real-time quantitative polymerase chain reaction (RT-qPCR). The results suggested that in BPH-1 cells, Cu B upregulated *MDM2* and *THBS1* gene expression levels. Of note, Cu B had no significant effect on the *TP53* gene level (Figure 4).

### 2.4. Cu B Upregulated TP53, MDM2, and THBS1 Protein Expressions in BPH-1 Cells

In order to comprehensively and systematically analyze the molecular mechanism of Cu B and the regulation of the p53 signaling pathway, we measured the expression of p53-related proteins using immunocytochemistry. We observed that Cu B upregulated TP53 and MDM2 protein expressions in BPH-1 cells (Figure 5). Considering that THBS1 is a secretory protein, we also measured the secretion of THBS1 in the cell culture supernatant, and the results suggested that Cu B significantly increased the secretion of THBS1 in BPH-1 cells (Figure 6).

### 2.5. Cu B Downregulated COX-2 Expression in BPH-1 Cells

Considering the importance of inflammatory factors in the pathogenesis of benign prostatic hyperplasia, we also evaluated the levels of COX-2 expression in the BPH-1 cell culture supernatant. Our findings showed that Cu B (50 nM) markedly decreased COX-2 expression in BPH-1 cells (Figure 6).

## 3. Discussion

The uncontrolled proliferation of BPH-1 epithelial cells contributes to the pathological process of prostate lesions [8]. Some studies have shown that natural products could inhibit BPH-1 cell proliferation and exert anti-BPH effects. For example, umbelliferone derived from coumarin could inhibit the proliferation and cell cycle progression of BPH-1 cells via the STAT3/E2F transcription factor 1 (E2F1) signaling axis [9]. In this study, we evaluated the anti-proliferative activity of Cu B in vitro using BPH-1 cell lines. The results highlighted that Cu B effectively inhibited prostate cell proliferation and induced cell apoptosis. We extrapolated that Cu B exerted an inhibitory effect on BPH-1 cells via the p53/MDM2 axis, which manifests as the up-modulation of p53, MDM2, and THBS1 expression levels and the downregulation of COX-2 protein expression in BPH-1 cells.

The *TP53* gene functions as a tumor suppressor and contributes to maintaining cell DNA integrity. Studies have shown that p53 is involved in multiple biological processes, including cell cycle arrest, apoptosis, and cell differentiation [10]. Mechanistically, p53 can be controlled by its negative regulator MDM2. The human *MDM2* gene is located on chromosome 12q 12.3-q15 and codes 491 amino acid residues, which form multiple functional binding sites [11]. The *MDM2* gene includes P1 and P2 promoters, and the P2 promoter is the negative regulator of *p53*. The *MDM2* N-terminal domain can bind to the *p53* transactivation domain and restrain *p53* transcriptional activation. Meanwhile, the *MDM2* C terminus RING domain exerts E3 ligase activity, which mediates p53 ubiquitination and degradation [12]. In our study, Cu B significantly augmented p53 protein expression but did not affect the TP53 transcriptional level in BPH-1 cells, underscoring that Cu B may not directly induce p53 transcriptional activation but participate in its translational regulation. The activation of p53 protein expression was involved in Cu B-induced BPH-1 cell apoptosis. Nevertheless, Cu B treatment upregulated the MDM2 protein in BPH-1 cells. The upregulation of MDM2 can be attributed to different cell types and pleiotropic functions of MDM2. On the one hand, MDM2 has been reported to have a higher basal expression in BPH-1 cells. Transient or stable MDM2 silencing slightly enhanced BPH-1 cell migration, and the downregulation of MDM2 in BPH-1 cells makes them insensitive to docetaxel treatment [13]. On the other hand, MDM2 is usually considered a negative regulator of the p53 protein, and many clinical trials were also dedicated to targeting p53-MDM2 interactions or inhibiting MDM2. However, no MDM2 inhibitor has been successfully launched. MDM2 may affect biological processes in a p53-independent manner. A disruptive study demonstrated that MDM2 regulates T cell STAT5 stability, T cell survival, and anti-tumor immunity. At the same time, the authors found that APG115, an inhibitor of the p53/MDM2 interaction, promoted p53 and MDM2 expression in mouse and human T cells and that APG115 could inhibit multiple types of tumor growth. That study indicated that MDM2 may have a crucial auxiliary role in the anticancer process [14]. Additionally, MDM2 may be conducive to inhibiting mitotic progression and cell proliferation [15,16]. Based on the aforementioned study, we considered that the up-modulation of MDM2 in BPH-1 cells induced by Cu B may be ascribed to, at least partly, the differential biochemical functions of MDM2 and different cell types, and the mentioned findings may underlie the molecular basis for a potential antiproliferative effect of Cu B on BPH-1 cells.

THBS1, a p53 downstream target molecule, has gained increased attention because of its ability to pleiotropically modulate a variety of secreted proteins, including TGF-β1, proteases, angiogenic growth factors, and frizzled-related protein (sFRP)-1 [17]. THBS1 expression was downregulated in glioblastoma multiforme (GBM), which may be implicated in THBS1 promoter methylation and transcriptional silencing [18]. In the tumor microenvironment, THBS1 has multifaceted roles in inhibiting angiogenesis and regulating antitumor immunity [19]. Enhancer of zeste homolog 2 (EZH2) directly targets THBS1, thus promoting neuroendocrine progression and angiogenesis in aggressive prostate cancer [20]. Moreover, histone deacetylase-2 (HDAC2) suppressed THBS1 expression, subsequently induced angiogenesis, and promoted prostate cancer progression mediated by beta-adrenergic signaling [21]. The mentioned evidence revealed that THBS1 functioned as a potential therapeutic target for cancerous malignancy or proliferative vascular diseases. In vitro studies have indicated that allicin, a natural product, induces breast cancer cell apoptosis and cycle arrest by modulating the p53 signaling-related protein expression of THBS1, alpha-1-B glycoprotein (A1BG), and tropomyosin alpha-4 chain (TPM4) [22]. In our study, we found that Cu B significantly upregulated the gene level of *THBS1* in BPH-1 cells. Having in mind that THBS1 is a secretory protein, we also examined the culture supernatant level of BPH-1 cells. The results highlighted that Cu B dramatically upregulated THBS1 protein expression, which indicated that THBS1 might play a partial role in the apoptotic response after Cu B treatment. Molecular mechanism studies are needed to shed light on the more enigmatic p53 signaling pathway, including the role of MDM2 and THBS1 in inhibiting the cell proliferation process. Consequently, it would be thought-provoking to further investigate whether Cu B treatment induced the functional crosstalk among p53, MDM2, and THBS1.

COX-2 is responsible for the conversion of arachidonic acid into prostaglandins (PGs). Various pro-inflammatory cytokines and microbial agents induce COX-2 secretion and modulate signal transduction of proliferation, apoptosis, inflammation, differentiation, and metastatic process. Therefore, COX-2 represents a potential molecular target for preventing and treating prostatic lesions. In vitro and in vivo model systems showed that COX-2 is involved in pro-oncogenic kinase PKCε-mediated prostatic adenocarcinoma, and COX-2 inhibitors selectively induced the overexpression of PKCε prostate epithelial cell apoptosis [23]. Some natural products have also been found to inhibit prostate hyperplasia by inhibiting the expression level of COX-2. For example, aescin derived from *Aesculus hippocastanum* seeds has been reported to inhibit BPH by decreasing the expression levels of inflammatory factors, including IL-1β, TNF-α, and COX-2 [24]. Additionally, Yongdamsagan-tang, a traditional herbal formula, could inhibit BPH-1 cell proliferation via cell-cycle arrest and the downregulation of COX-2 expression [25]. The Cucurbitaceae family has been associated with anti-inflammatory properties. Cucumis melo seeds containing cucurbitacin B, D, and E have shown inhibitory effects on COX-2. These seeds can be utilized for analgesic and anti-inflammatory treatment [26]. Cucurbitacin E effectively suppressed carrageenan-induced rat paw edema by inhibiting COX-2 expression [27]. In our study, Cu B decreased COX-2 expression in the BPH-1 cell culture supernatant. As a result, we posited that Cu B could pleiotropically modulate the secretion of COX-2, which enabled Cu B to inhibit prostate cell proliferation.

## 4. Materials and Methods

### 4.1. Chemicals

Cu B (Lot#23114, 99.91% pure) was derived from MedChemExpress (Shanghai, China). The Cell Counting Kit-8 (CCK-8), 4% paraformaldehyde fix solution, enhanced immunostaining permeabilization buffer, DAB horseradish peroxidase color development kit, QuickBlock™ blocking buffer for immunol staining, and QuickBlock™ primary and secondary antibody dilution buffers for immunohistochemistry were purchased from Beyotime Biotechnology (Shanghai, China). Annexin V-FITC/PI kits were from Yeasen (Shanghai, China).

### 4.2. Cell Culture

The BPH-1 cells were from YSRIBIO industrial co., LTD. (Shanghai, China). The BPH-1 cells were cultured in F-12K medium supplemented with 10% FBS and 1% P/S. All cells were cultured in an incubator containing 5% CO_2_ at 37 °C.

### 4.3. Cell Viability and Cell Morphology

Cells in the logarithmic growth phase were made into single-cell suspension with trypsin digestion and counted using a blood cell counting board. BPH-1 cells (4000 or 3000 per well) were inoculated in 96-well plates with six replicate wells for 24 h at 37 °C and 5% CO_2_ and then treated with vehicle (0.1% DMSO (*v*/*v*)), doxazosin (40 μM), or Cu B (12.5, 25, 50, 100, 200 nM) for 48 h and 72 h. Subsequently, 10 μL of CCK8 reagent was added to each well and incubated for 1–4 h at 37 °C. The optical density (OD) of each well was measured using a microplate reader at 450 nm (BioTek, Winooski, VT, USA). CCK-8 assays were performed in triplicate. The cell viability was calculated as Cell viability (%) = [(A_sample_ − A_blank_)/(A_vehicle_ − A_blank_)] ×100%, where A_sample_ is the optical density of wells with cells, CCK-8 reagent and Cu B solution, A_blank_ is the optical density of wells with medium and CCK-8 reagent but no cells, and A_vehicle_ is the optical density of wells with cells and CCK-8 reagent but no Cu B solution.

### 4.4. Annexin V-FITC/PI Staining

Annexin V-fluorescein isothiocyanate (FITC) and propidium iodide (PI) staining were utilized to analyze cell apoptosis. BPH-1 cells (2.5 × 10^5^ cells/well) were seeded in six-well plates for 24 h and then vehicle (0.1% DMSO), doxazosin (40 μM), or Cu B (12.5 nM, 25 nM, 50 nM,100 nM) was added for 48 h. Then, the cells were washed once with PBS, digested with trypsin, and centrifuged at 900 rpm for 3 min. The clear supernatant of cells was discarded and resuspended with 100 μL binding buffer, followed by incubation with 1 μL Annexin V-FITC and 2 μL PI staining solution. Apoptotic cells were incubated in darkness for 10–15 min at room temperature, followed by adding 400 μL binding buffer with sufficient mixing. All samples were assessed using flow cytometry within 1 h (BD Biosciences, Franklin Lakes, NJ, USA). The results were analyzed using BD FACSDiva software v8.0.1.

### 4.5. Real-Time Quantitative Polymerase Chain Reaction (RT-qPCR)

Real-time quantitative polymerase chain reaction (RT-qPCR) was used to measure the relative quantities of TP53, MDM2, and THBS1 mRNA levels between the control group and Cu B-treated group (12.5 nM, 25 nM, 50 nM) (*n* = 3). Total RNA from prostate cells was isolated with an RNAeasyTM kit (R0026, Beyotime Biotechnology, Shanghai, China). The concentration and purity of RNA samples were assessed using a spectrophotometer (ACT gene, Shanghai, China). cDNA was generated from 1 μg RNA on a PCR machine (Applied Biosystems, Shanghai, China) according to the procedures of the PrimeScript™RT Master Mix (Takara, Shanghai, China). Primer sequences were designed using premier primer 6, as shown in Table 1. In order to measure mRNA expression, RT-qPCR analysis was performed using TB Green^®^Fast qPCR Mix (Takara, Shanghai, China). The mRNA levels were quantified using a Roche LC480 instrument (Roche, Basel, Switzerland). The relative gene expression was analyzed using the 2^−∆∆Ct^ method.

### 4.6. Immunocytochemistry

The BPH-1 cells (5000 cells/well) were seeded on the cell culture slide (BD Biosciences, 354104, Franklin Lakes, NJ, USA). Cells were allowed to adhere overnight and were then incubated with vehicle (0.1% DMSO) or cucurbitacin B (12.5 nM, 25 nM, 50 nM) for 24 h. Then, the slides were washed three times with phosphate-buffered saline (PBS) and fixed with 4% paraformaldehyde (PFA), followed by permeabilizing with enhanced immunostaining permeabilization buffer and blocking with QuickBlock™ blocking buffer for 20 min at room temperature. The cells were incubated overnight at 4 °C with TP53 primary antibody (1:100, AF0255, Beyotime, Shanghai, China) and MDM2 primary antibody (1:100, D160611, Sangon Biotech, Shanghai, China). Negative controls omitted primary antibodies. After washing with PBS, cells were incubated with HRP-conjugated goat anti-mouse IgG (1:200, D110087, Sangon Biotech, Shanghai, China) or HRP-conjugated goat anti-rabbit IgG (1:200, D110058, Sangon Biotech, Shanghai, China) for 1 h at room temperature. Subsequently, the cells were washed three times with PBS and treated with a DAB staining solution. The samples were observed and analyzed using an inverted microscope (AOSVI, Shengzhen, China). Three cell sections were made in each group, and three microscopic fields were randomly selected from each section, the semi-quantification expressions of TP53 and MDM2 were evaluated using Image J 1.8.0 software.

### 4.7. Enzyme-Linked Immunosorbent Assay (ELISA)

BPH-1 (3 × 10^5^ cells/well) was inoculated in 6-well plates, cultured for 24 h, and treated with Cu B (12.5 nM, 25 nM, 50 nM) for 48 h. Subsequently, BPH-1 cell supernatants were centrifuged for collection. COX-2 and THBS1 concentrations were measured using human ELISA Kits (COX-2, JL19470, Jianglaibio, Shanghai, China; THBS1, JL11889, Jianglaibio, Shanghai, China) according to the manufacturer’s recommendations. The COX-2 and THBS1 protein levels in the BPH-1 cells were quantified using an enzyme-linked immunosorbent analyzed at 450 nm (BioTek, Winooski, VT, USA).

### 4.8. Statistical Analysis

The data are presented as mean value ± standard deviation (SD). All experiments were carried out in three biological replicates. The data were analyzed using one-way ANOVA statistics and the least significant difference post hoc test or Dunnett’s post hoc test using SPSS version 26.0 software. A *p*-value < 0.05 was considered statistically significant. Graphical data were visualized using GraphPad Prism 8.0 (GraphPad Software, San Diego, CA, USA).

## 5. Conclusions

Taken together, this study revealed that Cu B effectively inhibited the proliferation of BPH-1 cells and was related to p53 signaling transduction. The molecular mechanism research indicated that Cu B affected p53 and the downstream gene or protein expressions of THBS1 and MDM2. In addition, Cu B could downregulate COX-2 protein expression. Thus, our findings suggest that Cu B may inhibit the proliferation of prostate cells and induce cell apoptosis. However, whether there is a crosstalk between p53 and other signaling pathways in Cu B-induced cell apoptosis awaits further investigation.

## Figures and Tables

**Figure 1 ijms-25-00442-f001:**
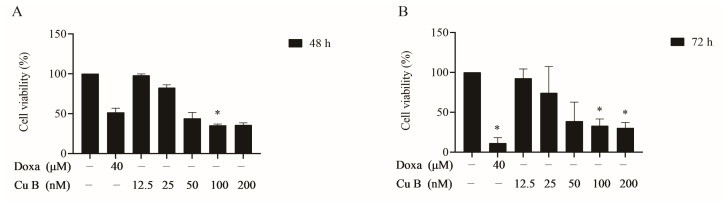
Effect of Cu B on the cell viability of BPH-1 cells based on CCK-8 results. Prostate cells were treated with vehicle (0.1% DMSO), doxazosin (40 μM), or Cu B (12.5 nM, 25 nM, 50 nM, 100 nM, 200 nM) for 48 h (**A**) and 72 h (**B**). Cu B: cucurbitacin B; Doxa: doxazosin; CCK-8: counting kit-8; BPH-1: benign prostatic hyperplasia epithelial cell line; DMSO: dimethyl sulfoxide. The data are presented as mean ± standard deviation (SD) and were analyzed using one-way analysis of variance (ANOVA) followed by a least significant difference post hoc test or Dunnett’s post hoc test (*n* = 3, * *p* < 0.05, compared with the control).

**Figure 2 ijms-25-00442-f002:**
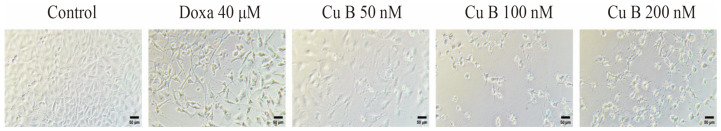
Effect of Cu B (50–200 nM) and doxazosin (40 μM) on the cellular morphology of BPH-1 cells treated for 48 h (100×). Cu B: cucurbitacin B; Doxa: doxazosin; BPH-1: benign prostatic hyperplasia epithelial cell line. Scale bar = 50 μm.

**Figure 3 ijms-25-00442-f003:**
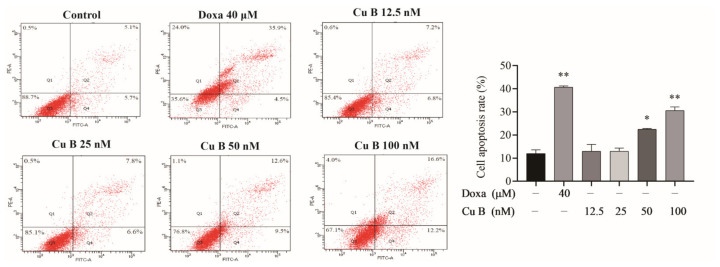
Cu B and Doxa induced the apoptosis of BPH-1 cells based on the results of the flow cytometry analysis. Apoptosis of BPH-1 cells that were treated with Cu B and Doxa for 48 h. Cu B: cucurbitacin B; Doxa: doxazosin; BPH-1: benign prostatic hyperplasia epithelial cell line. The data are presented as mean ± SD and were analyzed using one-way ANOVA followed by a least significant difference post hoc test or Dunnett’s post hoc test (*n* = 3, * *p* < 0.05, ** *p* < 0.01, compared with the control).

**Figure 4 ijms-25-00442-f004:**
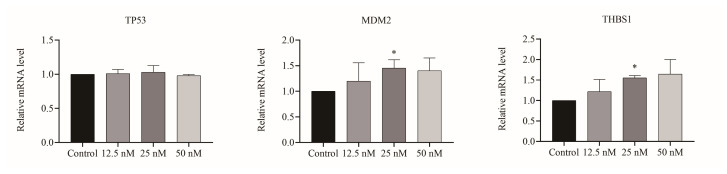
Cu B regulated the expression of genes in BPH-1 cells based on the RT-qPCR results. The gene expression level of *TP53*, *MDM2*, and *THBS1* in Cu B-treated cells for 48 h. Cu B: cucurbitacin B; THBS1: thrombospondin 1; BPH-1: benign prostatic hyperplasia epithelial cell line; RT-qPCR: real-time quantitative polymerase chain reaction. The data are presented as mean ± SD and were analyzed using one-way ANOVA followed by a least significant difference post hoc test or Dunnett’s post hoc test (*n* = 3, * *p* < 0.05, compared with the control).

**Figure 5 ijms-25-00442-f005:**
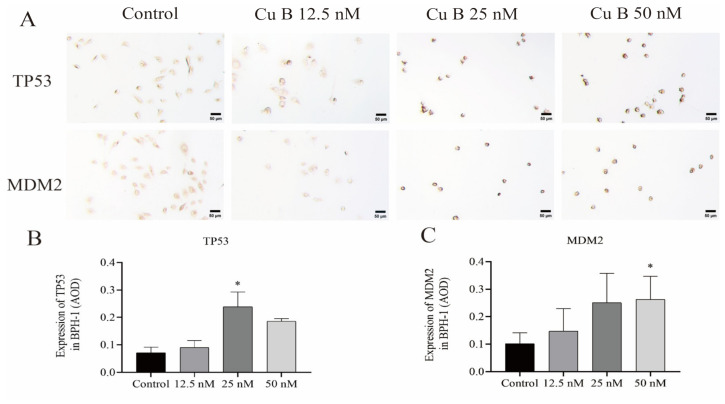
Effect of Cu B on the protein expressions of TP53 and MDM2 in BPH-1 cells. (**A**) Immunocytochemistry images of TP53 and MDM2 in BPH-1 cells (400×). (**B**) Effect of Cu B on TP53 expression. (**C**) Effect of Cu B on MDM2 expression. Cu B: cucurbitacin B; BPH-1: benign prostatic hyperplasia epithelial cell line; AOD: average optical density. Scale bar = 50 μm. The data are presented as mean ± SD and were analyzed using one-way ANOVA followed by a least significant difference post hoc test or Dunnett’s post hoc test (*n* = 3, * *p* < 0.05, compared with the control).

**Figure 6 ijms-25-00442-f006:**
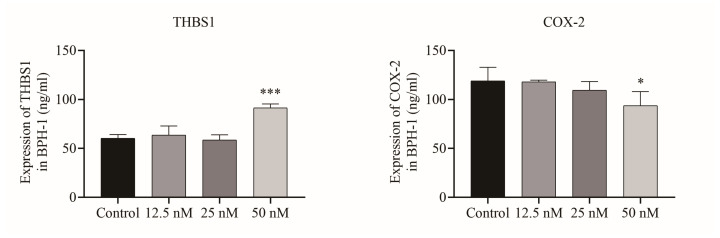
ELISA analysis results of THBS1 and COX-2 in the BPH-1 cell culture supernatant. BPH-1 cells were treated with vehicle (0.1% DMSO) and Cu B (12.5 nM, 25 nM, 50 nM) for 48 h. ELISA: enzyme-linked immunosorbent assay; THBS1: thrombospondin 1; DMSO: dimethyl sulfoxide; Cu B: cucurbitacin B; BPH-1: benign prostatic hyperplasia epithelial cell line. The data are presented as mean ± SD and were analyzed using one-way ANOVA followed by a least significant difference post hoc test or Dunnett’s post hoc test (*n* = 3, * *p* < 0.05, *** *p* < 0.001, compared with the control).

**Table 1 ijms-25-00442-t001:** Primer sequences for RT-qPCR.

Genes	Forward Primer (5′-3′)	Reverse Primer (5′-3′)
*TP53*	CACTAAGCGAGCACTGCCCAACA	GCCTCATTCAGCTCTCGGAACATCT
*MDM2*	TTGGCGTGCCAAGCTTCTCTGTG	ACCTGAGTCCGATGATTCCTGCTGA
*THBS1*	ATGGAGAATGCTGTCCTCGCTGTTG	CGGTTGTTGAGGCTATCGCAGGAG
*GAPDH*	CAGGAGGCATTGCTGATGAT	GAAGGCTGGGGCTCATTT

## Data Availability

The data presented in this study are available on request from the corresponding author.

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
