# Peer review of "The Antiproliferative and Proapoptotic Effects of Cucurbitacin B on BPH-1 Cells via the p53/MDM2 Axis"

_ijms, 2023, doi:10.3390/ijms25010442_

Round 1
Reviewer 1 Report
Comments and Suggestions for Authors
In the manuscript titled “The antiproliferative and proapoptotic effects of cucurbitacin B on BPH-1 cells through p53/MDM2 axis”, the authors report the inhibition of cucurbitacin B on benign prostatic hyperplasia epithelial cell line (BPH-1) studying the molecular mechanism, in which is involved the upregulation expression of p53 and MDM2. Furthermore, a molecular docking calculation on COX-2 enzyme was performed. The study is interesting, but the authors have limited the molecular docking study only on COX-2, although the experimental molecular mechanism involves also p53 and MDM2. In addition, the molecular docking on COX-2 has many issues:
1. The molecular structure PDB-ID: 5F19, is the acetylated form at two serine residues of COX-2, obtained by aspirin treatment which is no longer present on the active site. The absence of an inhibitor during the crystallization process doesn’t allow to form the pocket in the active site, so the structure it may have “collapsed” and not allow to obtain realistic docking results, unless you also carry out a molecular dynamics calculation in the presence of the molecule under investigation. Furthermore, it is not possible to establish where is the catalytic pocket in the protein structure without an inhibitor. The authors would have had to choose a COX-2 enzyme with the presence of an inhibitor as PDB-ID: 5KIR. Please redone docking calculations on these enzymes, performing a preliminary validation by re-docking the original X-ray structure ligands, but not starting from the original ligand position in the pocket, in order to avoid artefacts.
2. In the experimental part regarding the docking calculation (paragraph 4.8) it is not well explained how the authors have elaborate the structure of native enzyme: the not essential ligands as EDO, NAG, AKR, BOG, Protoporphyrin IX containing Co and water molecules were removed?
3. The dimension and the coordinate of the box used in the calculation need to be reported.
4. Have the authors used AutoDock or AutoDock Vina to perform the calculation? The authors need to report the most important parameters used in the calculation and not write “standard docking process”.
5. Have the authors minimized the raw structure of cucurbitacin B taken in which format from Pubchem Database? Please provide to minimize the structure and report a figure containing the minimized structure.
6. The authors should to study by docking calculation the effect of cucurbitacin B on MDM2 by using a structure of MDM2 with low molecular weight inhibitor as PDB-ID: 4ZGK or 3TU1.
Minor issues are:
1. The figures resolution is low and it needs to be improved.
2. Please, insert a table containing the energy score values and the interactions types (not only hydrogen bonds as showed in figure 7) involved with amino acidic residues for cucurbitacin B and for the original ligand deriving from X-ray structure after docking calculation, comparing the differences.
In conclusion, the manuscript can be accepted after major revision performing molecular docking calculation on COX-2 with PDB-ID: 5KIR and on MDM2 with PDB-ID 4ZGK or 3TU1, giving in the experimental part all the details used for the calculation and adding to paragraphs Results and Discussion the obtained new results. An alternative is to remove the docking section, so it can be published in this form, improving only figures resolution.
Reviewer 2 Report
Comments and Suggestions for Authors
The manuscript entitled The antiproliferative and proapoptotic effects of cucurbitacin B on BPH-1 cells through p53/MDM2 axis summarizes the in vitro effects of a plant-derived compound that is a potential natural drug in the treatment of benign prostate hyperplasia.
The experimental concept is well established and clear and the results can be considered novelties. The topic is very relevant due to the focus on a global health problem and the authors clarified important mechanistic aspects of cucurbitacin B.
The methodology is versatile and sophisticated, simple colorimetric viability assays and molecular methods are also used to assess the possible mechanism of action of the tested triterpenoid compound. My question regarding the methodology is how many independent experiments were done of each experimental method?
The figures in the downloadable version of the article are in non-publishable quality, all of them are blurred. Please, improve the resolution of the figures, especially the microscopy photos. Unfortunately, the microscopic morphological changes and immunocytochemistry results cannot be objectively reviewed in this form.
The style of the language in the article is good and there are only a few wrong spellings or missing spaces between letters. Missing space between the last letter and the square brackets of the reference numbers are repetitively occurring (e.g. 34. row receptor[1]). Please, correct it in all cases.
Row 213-214: The sentence is inaccurate because it is written that the MDM2 gene contains amino acids, however, it is only coding amino acids. Please, correct it.
In the introduction part, one paragraph is dedicated to p53 functions that are related to androgen receptor functions, however, the tested cell line is AR-negative. Please, clarify why You found it important to emphasize the relation of p53 to AR.
In the Discussion (rows 265-269), COX-2 functions are also linked to testosterone-induced processes. How can it be related to the AR-negative BPH-1 cells? Please, complete this section with the explanation.
Please, check the official abbreviations of the cited journals. In the references, there are a few inaccuracies (e.g. journal names should be written with capitals as the first letter (ref. 5 and 14)
Please, dedicate a new paragraph to the explanation in the Discussion part to the finding that p53 and MDM2 are both activated by Cu B. If MDM2 negatively regulates p53, what is the benefit of MDM2 activation (in mRNA and protein levels)?
Comments on the Quality of English Language
The quality of the English language fits the requirements of scientific writing. Minor grammar, spelling, or punctuation problems need to be corrected.
Round 2
Reviewer 1 Report
Comments and Suggestions for Authors
Removing the docking section allows the paper to be accepted in this form.